

# Swiss Early Instrumental Meteorological Measurements

Lucas Pfister[1,2], Franziska Hupfer[1,2], Yuri Brugnara[1,2], Lukas Munz[1,2], Leonie Villiger[1,2], Lukas Meyer[1,2], Mikhaël Schwander[1,2], Francesco Alessandro Isotta[3], Christian Rohr[1,4] and Stefan Brönnimann[1,2]

[1] Oeschger Centre for Climate Change Research, University of Bern, Switzerland

[2] Institute of Geography, University of Bern, Switzerland

[3] Federal Office of Meteorology and Climatology MeteoSwiss, Zurich, Switzerland

[4] Institute of History, University of Bern, Switzerland

*Correspondence to*: Lucas Pfister (lucas.pfister@giub.unibe.ch)

**Abstract.** Decadal variability of weather and its extremes is still poorly understood. This is partly due to the shortness of records, which, for many parts of the world, only allow studies of 20[th] century weather. However, the 18[th] and early 19[th] century have seen some pronounced climatic variations, with equally pronounced impacts on the environment and society. Considerable amounts of weather data are available even for that time, but have not yet been digitised. Given the recent progress in quantitative reconstruction of sub-daily weather, such data could form the basis of weather reconstructions. In Switzerland, measurements before 1864 (the start of the national network) have never been systematically compiled except for three prominent series (Geneva, Basel, Great St. Bernard Pass). Here we provide an overview of early instrumental meteorological measurements in Switzerland resulting from an archive survey. Our inventory encompasses 335 entries from 206 locations, providing an estimated 3640 station years and reaching back to the early 18[th] century. Most of the data sheets have been photographed and a considerable fraction is undergoing digitisation. This paper accompanies the online publication of the imaged data series and metadata. We provide a detailed inventory of the series, discuss their historical context and provide the photographed data sheets. We demonstrate their usefulness on behalf of two historical cases and show how they complement the existing series in Europe. If similar searches in other countries yield similarly rich results, an extension of daily weather reconstructions for Europe back to the 1760s is possible.

## 1. Introduction

While decadal-to-multidecadal climate variability has long become a major research focus in recent years, decadal variability in weather is still poorly understood. This is partly due to the fact that the observed record covers only an insufficient number and variety of decadal climatic anomalies and changes. Thus, there is a need to extend the record back in time. A monumental effort to draw historical daily weather maps was undertaken in the 1980s (Kington, 1988). A suite of recent, numerical approaches has targeted the reconstruction of weather based on early instrumental meteorological measurements, comprising data assimilation (e.g., the "Twentieth Century Reanalysis" 20CR, Compo et al., 2011; or the coupled European reanalysis of the 20[th] century CERA-20C, Laloyaux et al., 2018), daily weather-type reconstruction (Schwander et al., 2017; Delaygue et al., 2018) or analogue approaches (Yiou et al., 2014; Flückiger et al., 2017). These approaches allow, at least partly, addressing changes in atmospheric dynamics. 20CR reaches back to the mid-19[th] century, i.e., the start of national weather services. Can we analyse weather variability even further back in time? In this paper we show, for the case of Switzerland, that rich, hitherto unknown data sources can still be found today in archives.

The 18[th] and early 19[th] century have seen pronounced climatic variations in Central Europe such as a much discussed warm and dry period around 1800 (e.g., Frank et al., 2007, Böhm et al., 2010), which was followed by an extremely cold period during the early 19[th] century (Brönnimann, 2015). Furthermore, a period of increased flood frequency occurred in the 19[th]



century in Switzerland (Pfister, 1999; Schmocker-Fackel and Naef, 2010; see also Brönnimann et al., 2019). These climatic
events had pronounced impacts on environment and society (e.g., Summermatter, 2005, Stucki et al., 2018), which makes
them interesting examples to study. Present climate risk management can make use of historical weather data and historical
documents (Brönnimann et al., 2018a).

In Switzerland, a national meteorological network was initiated in December 1863 and maintained by the "Schweizerische
Naturforschende Gesellschaft" (Swiss Society of Natural Sciences) and later MeteoSwiss. These data are well documented
and a considerable fraction has been digitised and is electronically available (e.g., Begert et al., 2005). Three prominent
series that reach further back than 1864 have been re-evaluated (on the basis of daily or monthly mean values) in the 1950s
(Bider et al., 1959; Schüepp, 1961), namely Geneva, Basel (both back to the mid-18[th] century), and Great St. Bernard Pass
(back to 1819). These series have been important in climate science and for climate reconstruction, for which monthly re-
solved temperature series form the basis. They are also incorporated in the Austria-led database HISTALP (Auer et al.,
2007). We have recently re-digitised these three long records based on original values rather than daily means (Füllemann et
al., 2012), but not the full set of records that would possibly be available. In fact, measurements before 1864 have not been
systematically compiled recently. Starting in 2005, MeteoSwiss registered its archive collections (in collaboration with the
company GRAD GIS/Thomas Specker) and transferred them to the Swiss Federal Archives in Bern or the State Archives of
the canton of Zurich (Fonds „Naturforschende Gesellschaft in Zürich"). This made many Swiss records easier to access.

Already in 1864 Rudolf Wolf, director of the "Schweizerische Meteorologische Zentralanstalt" (Swiss Central Meteorolo-
gical Agency), gathered information about meteorological records in Switzerland dating back to the 16[th] century and pointed
out their scientific value (MZA, 1864). The aim was to make several of these records accessible, but only few early instru-
mental data series were later published in some of the first Swiss meteorological annals in the 19[th] century. Unfortunately,
they were mostly published in the form of daily mean values (within the EU project EMULATE, these daily data were digit-
ised back to 1850 for Geneva, Zurich/Uetliberg and Bern). In the 1920s, Billwiller (1927) published a more comprehensive
inventory of Swiss meteorological records prior to 1864, compiling the available information on the series (but not the data).
Individual studies have addressed some of the early records (e.g. Gisler, 1983). Gisler (1983, 1985) analysed instrumental
series from Schaffhausen and Zurich, however, only based on monthly mean values. Some of the shorter series have been
analysed as an object of historical studies (e.g. Burri and Zenhäusern, 2009), but the original data have not been digitised.
This is particularly true for instrumental measurements while precipitation observations were sometimes digitised and incor-
porated into databases (e.g. EURO-CLIMHIST; see also Gimmi et al., 2007). Although meteorological networks in Switzer-
land in the mid-19[th] century have been analysed by historians (e.g., Pfister, 1975; Hupfer, 2015; Hupfer, 2019), no attempts
have been undertaken to compile and digitise the data.

Here we report the results from a systematic survey of early instrumental meteorological measurements in Switzerland.
Archive work brought to light more than 300 station records prior to 1864, some reaching back to the early 18[th] century. This
paper accompanies the online publication of the photographed data sheets and metadata. A subsequent paper will describe
the digitised records. Note that Switzerland here serves as an example of what could be found elsewhere in Europe.

The paper is organised as follows. Section 2 provides an overview of the history of meteorological measurement in Switzer-
land from the 18[th] century to the start of the national weather service. Section 3 describes the archive work and the resulting
inventory. In Section 4 we demonstrate the usefulness of the photographed data sheets on behalf of two examples. Conclu-
sions are drawn in Section 5.



## 2. History of Meteorological Measurements in Switzerland

In the mid-17th century weather observation started to change from being qualitative to largely quantitative. The instrumental measurements were a trend that originated in the "quantifying spirit" of the Enlightenment, making instrumentation a new priority for scholars in Europe (Frängsmyr et al., 1990; Bourguet et al., 2002). However, instrumental measurements have
only gradually replaced qualitative assessments of the weather. The instrumental method did not reach its supremacy in meteorological observation until the 19th century (Janković, 2001). The quantification process interacted with the development of meteorological instruments. The first thermometers were built in the early 1600s, the first barometers some decades later, with contributions from several well-known naturalists such as Descartes (Middleton, 1969; Golinski, 1999). Also hygrometers and instruments for measuring precipitation, wind direction, and wind force came into use in the 17th century. Sub-
sequently, these devices underwent technical changes, thus being an important part of the history of meteorological data.

Scholars took along meteorological instruments on travel, observing on mountains and other special locations that they thought deserved attention. Some also conducted regular instrumental measurements in their home towns. In Switzerland, Johann Jakob Scheuchzer (1672-1733), a physician and professor of mathematics in Zurich, well-connected within the scholarly community of his time, made daily observations of atmospheric pressure, temperature, and precipitation. Starting in
1708, his records (with some interruptions until 1731) are the oldest instrumental series on the territory of today's Switzerland. Although the original diary is lost, many of Scheuchzer's results are known thanks to his publications in journals (Pfister, 1984). Scheuchzer's interest in weather data was part of his larger program to collect all sorts of information on Swiss and Alpine natural history (Pfister, 1975; Boscani Leoni, 2013; Boscani Leoni, 2016). His efforts to get meteorological observations from peripheral regions had little success in general. Nevertheless, he managed to arrange measurements
on St. Gotthard Pass: The Capuchin hospice agreed to read every day the barometer that Scheuchzer had installed there (Fischer, 1973). This allowed for corresponding observations between Zurich and St. Gotthard Pass (1728-1730). Observational programs in order to compare meteorological measurements were also conceived by scientific societies or academies. The earliest example is the Florentine Accademia del Cimento, which had sent thermometers to several places in Europe shortly after its foundation in 1657 (Hellmann, 1901; Hellmann, 1914). These data have been digitised and evaluated
(Camuffo and Bertolin, 2012).

From the early 18th century on, attempts were made to publish compilations of data from different locations. When the Wrocław physician Johann Kanold invited other savants to submit their records for publication, Scheuchzer provided the Zurich data (1718-1726, see Steiger, 1933). Between 1718 and 1730, Kanold collected and printed meteorological information from more than 20 European cities in his *Breslauer Sammlung* (Hellmann, 1926; Feldman, 1990; Lüdecke, 2010). The
Royal Society in London carried out a similar project, initiated in 1723 by its Secretary James Jurin who issued an invitation to potential observers. During circa ten years, the Royal Society published Jurin's collected responses in its journal (Hellmann, 1914; Daston, 2008). Such undertakings illustrate the growing interest of 18th-century scholars to compare empirical details. As a result, exchanges across the political borders of Europe were stimulated.

In the second half of the 18th century, many projects were initiated to coordinate and publish meteorological observations,
most of them by scientific, economic or agricultural societies. Among the organisations with a meteorological program was the "Oekonomische Gesellschaft Bern" (Economic Society of Bern). Soon after its foundation in 1759, the society equipped several stations in different regions with thermometers and barometers (OeGB, 1762; Pfister, 1975). In the following decade, three to eight locations sent in measurements, of which monthly means and extremes were printed in the society's journal. This observation network set up by the "Oekonomische Gesellschaft" was small – regarding the number of stations and the
geographic reach (all stations were on the territory of the Republic of Bern, including its Vaudois territories and those ruled in common with Fribourg). Identically equipped stations represented though an innovative method initiated by such net-





works, and soon followed also on larger scale. The best-known 18[th]-century initiative to establish an extended network of observers is the Mannheim-based Palatine Meteorological Society (Societas Meteorologica Palatina, see Cassidy, 1985; Kington, 1988; Wege and Winkler, 2005; Lüdecke, 2010). It was a state-subsidised project: the elector of Palatine and of Bavaria,

Karl Theodor, provided the society's funds. Under his patronage, the court priest Johann Jakob Hemmer organised a network of over 30 stations in various parts of Europe. He also set up a station in Greenland and one in North America. All observers received a set of calibrated instruments (one barometer, two thermometers and one hygrometer) together with instructions on observational procedures (e.g., the subdaily observation times were fixed for 07 h, 14 h and 21 h). The registers were dispatched to Mannheim for publication in the society's journal called "Ephemerides" (twelve volumes 1781-1792). On the ter-

ritory of today's Switzerland, two observers participated in the Palatine Society's network: the pastor and librarian Jean Senebier in Geneva and the Capuchin monk Onuphrius on the St. Gotthard Pass, succeeded by fellow monks (Billwiller, 1927; Pfister, 1984; Grenon, 2010).

The ambitious plan of standardising instruments and observing practices within these early networks of meteorological stations strived for improving the comparability of the measurements. Given the high individuality of instruments in the 18[th]

century, it was extremely difficult to compare the observations (Daston, 2008). In this context, standardisation became an important issue. The participants of the late-18[th] and early 19[th]-century networks observed with more precise instruments, compared to most of the ones engaged outside coordinated projects. Also, they usually observed with more regularity (often subdaily at predetermined times) and submitted their tables to coordinators, who often published them in summaries or – less frequently – as full records. The growing amount of printed meteorological data was influenced by the expanding production

of scholarly journals in the 18[th] century (for scholarly media culture, see Holenstein et al., 2013). By defining publishing criteria for meteorological observations, networks had a standardising effect. However, the affiliation to a network did not necessarily guarantee high quality. Coordinators only had limited control over their observers. Many of them worked as volunteers and did not always follow their coordinators' guidelines that were often only poorly explained. Within a single network of stations, observations could vary significantly in reliability. Furthermore, not all networks had the resources to distribute

instruments, which limited the degree of standardisation.

Meteorological networks founded until the middle of the 19[th] century (before the era of official state observing systems) were all transitory. They seldom endured for more than 20 years (Edwards, 2010). Some never aimed at long-term series; others sought permanence but faced problems in resources. In the case of the Palatine society, the project declined after the death of its first coordinator. The French Revolutionary Wars further contributed to the disintegration of the network. The last annual

report was printed in 1795, 15 years after the promising start. The above-mentioned network of the "Oekonomische Gesellschaft Bern" was not a long-lasting one either: When the society as a whole became quite inactive for some time, its meteorological project was abandoned after ten years of existence (1760-1770). Most networks had a high fluctuation of observers: only a part of them pursued their measurements with tenacity over many years. The difficulties to build permanent structures persisted in the first half of the 19[th] century – despite the many endeavours to create and maintain networks with regional or

cross-border dimensions.

Within the territory of today's Switzerland, the early initiative of the "Oekonomische Gesellschaft Bern" was followed by several attempts of observational network building: In the 1810s, the "Naturforschende Gesellschaft" (Society of Natural Sciences) in Aarau sent barometers to circa ten observers in different European cities, but did not receive enough useful registers to publish a compilation (Hefty-Gysi, 1953). Some decades later, in the 1850s, the society set up a network inside the

canton of Aargau. However, most of the 22 stations stopped observations after less than three years (Hartmann, 1911). The network of the "Thurgauische Naturforschende Gesellschaft" (Society of Natural Sciences of Thurgau) was short-living as well (starting in 1855, see Bürgi, 2004). In the canton of Grisons, many members of the cantonal "Naturforschende Gesell-




schaft Graubündens" (Society of Natural Sciences of Grisons) participated in the observational system that Christian Gregor Brügger, a student, later secondary school teacher and natural history researcher, had initiated in 1856 (Hupfer, 2015). After
Brügger had left the region in 1859, the network lacked supervision and started to fall apart (although data were still collected). Some stations were incorporated into the national network starting in 1863. In the French-speaking part of Switzerland, the "Société des sciences naturelles de Neuchâtel" (Society of Natural Sciences of Neuchâtel) initiated a network in 1856 comprising six stations in the canton of Neuchâtel (Kopp, 1856). From these stations, however, only two persisted until the integration into the national network. Another, better controlled network was operated by the Observatory in Bern, which
had equipped eight stations in cooperation with the "Naturforschende Gesellschaft in Bern" (Society of Natural Sciences of Bern) and the cantonal government (starting in 1860, see Wild, 1860).

Besides these regional projects, the Swiss Society of Natural Sciences (established in 1815, today Swiss Academy of Sciences SCNAT) aimed at a network at national level. However, its first network with twelve stations, conceptualised by the Geneva observatory director Marc-Auguste Pictet in 1823, faced problems similar to those of other initiatives. It lacked a
financed coordination centre, was confronted with high publication costs and a considerable drop-off rate amongst its observers. Finally, the society abandoned its project in 1836 and printed only the means and extremes of three stations (Basel 1826-1836, Bern 1826-1836, St. Gallen 1827-1832, see Merian et al., 1838). The society was to have more success with its second attempt, undertaken in the 1860s, with support from the Swiss Federal State.

In the period of repeated efforts for expanded and durable observational enterprises, there were far more persons involved
than just those belonging to networks. Many observed as individuals, some on behalf of institutions such as universities or monasteries. Observations provided by institutions were often independent from the success or failure of meteorological networks. The Geneva observatory, for example, continued its measurements (started in 1772) regardless of whether it currently participated in a network or not (Gautier, 1843). Other long series evolved in Basel where, among others, the university professors Johann Jakob d'Annone and Peter Merian observed for almost fifty years each (1755-1804 and 1826-1874 respect-
ively, see Riggenbach, 1892; Bider et al., 1958). Especially Merian had an established status within the meteorological research. However, the qualification of the 18th- and 19th-century observers ranged wide, spanning from university scholars as Merian to persons without any scientific training. In the 18th century, thermometers and barometers became increasingly affordable, leading to a wider public use (Golinski, 2007). This diversity limits the possibility of generalisations about the observers.

The interest in meteorological measurements derived from both theoretical and practical goals. Most observers saw their systematic observations as a contribution to a scientific understanding of weather and climate (see for example Pictet, 1780). They hoped that their records might eventually show regularities that would allow finding natural laws or discerning correlations with celestial motions (for the Palatine Society's goals see: Cassidy, 1985, and more general: Daston, 2008). Although the observation networks could not provide current weather information before the widespread introduction of the telegraph
(invented in 1837), many organisers expected that their long-term data would help to predict the weather (Feldman, 1990). Others, particularly the "Oekonomische Gesellschaft Bern", were interested in the impact of atmospheric conditions on plants, hoping to derive practical measures in their effort to improve agricultural methods (Carrard, 1763; Pfister, 1975). Another source of interest were medical concerns, relating weather conditions to diseases, plagues or mortality (Fleming, 1990).

Whereas organisers of networks often explicitly outlined their motivation, little is known about the reasons for which indi-
viduals (connected to a network or not) started observations. The few testimonies reveal a desire to learn and to support knowledge production, often with a focus on their locality or region (Janković, 2001; Hupfer, 2015). In most cases, meteorological records were made by literate persons. However, well-off and almost exclusively masculine observers were often supported by household members who replaced them during their absences. Many observers not only monitored the weather,



but were also involved in other investigations of the nature such as phenological observations and botanical classifications. Some of them were well-known botanists, e.g., Laurent Garcin (1683-1752) or Abraham Gagnebin (1707-1800). A remarkable number of meteorological observers in 18th- and 19th-century Switzerland belonged to the liberal elite, among them the famous politician and author Heinrich Zschokke (1771-1848). Moreover, members of clergy formed an important group among the observers. When pastors were moved to another church, they often took their instruments with them, e.g., Rudolf Ludwig Fankhauser (1796-1886) who carried out measurements in three different parishes in the canton of Bern. Clergymen, physicians, pharmacists, lawyers or magistrates were not only typical meteorological observers, but were in general strongly represented in Europe's natural history community. In the pre-twentieth-century, this community was not limited to professional researchers, but included many non-academics.

In 1864, an important organisational transformation in meteorological measurement took place in Switzerland: a national observation network was started, organised by the "Schweizerische Naturforschende Gesellschaft" (Swiss Society of Natural Sciences) and supported by the Swiss Federal State (established in 1848). State-funding provided the financial stability that had lacked to the many previous attempts. The network was still a system of volunteer observers. However, its coordination was professionalised in form of an administrative centre, the "Schweizerische Meteorologische Zentralanstalt" (Swiss Central Meteorological Agency, later MeteoSwiss). This state-supported institution pushed standardisation of both observers and instruments and adapted international agreements about the recording and communication of data (Hupfer, 2017; Hupfer, 2019; on internationalisation: Edwards, 2010). After a few years, in 1881, the meteorological agency became an official state institution, responsible of climate monitoring and of synoptic weather forecasting based on international telegraphic data exchange.

The national observation networks that have emerged since the middle of the 19th century still form the basis of climate monitoring today. Compared to the 18th- and early 19th-century organisational structures, these state-founded networks can be considered as quite stable. Over the long term, measuring practices have developed towards an increased standardisation. However, the history of meteorological measurements is not only a success story, but also a history of many discontinued projects and unfulfilled expectations regarding both theoretical and practical aspects. Despite the less developed standards of the pre-1864 period, the inventoried series are informative for past weather events. What presents us with a challenge is the fact that measurement conditions were often not systematically documented. Therefore it is necessary to not only make the data available but also to examine the multiple contexts from which these observations originate.

### 3. Inventory

Our archive search started from the above mentioned previous compilations, most of which were published more than 100 years ago. Furthermore, we consulted the cantonal societies of natural sciences, checked their websites and conducted an online search targeted at digitally available journals and publications of these societies (Ephemerides, journals of the natural sciences societies available at e-periodica). In a next step, digital libraries were searched for meteorological data (e.g., Munich Digitization Center MDZ, Google Books, e-periodica, e-manuscripta and others), as well as publications of MeteoSwiss (Supplement Volumes of Annals of MeteoSwiss). Moreover, we consulted a number of libraries and archives (Swiss Federal Archives, Cantonal Archive Aargau, Cantonal Library Aargau, University Library of Basel, Burgerbibliothek of Berne, University Library of Bern, Archive of the Monastery of Einsiedeln, Cantonal/University Library Fribourg, Cantonal Archive and Library of Geneva, Cantonal Library/Archive Grisons, University Library of Neuchatel, City Archive Schaffhausen, Cantonal Library Vadiana St. Gallen, Cantonal Archive Vaud, Cantonal/University Library Lausanne, Cantonal Archive Zurich, City Library Zurich) and enquired many more libraries and archives about meteorological data in their holdings. Two examples of data sheets from the Swiss Federal Archives are displayed in Fig. 1.



From the known meteorological records, the vast majority could be located (see Fig. 2). The archive work revealed that
many more series exist than the ones known so far. If original records could be found (the majority of the series), the data
sheets were photographed. For certain series, transcripts and printed versions were photographed instead of missing original
manuscripts. In some cases, we could not locate the data but found information about the station or observer, or found data
only in statistical form and not as raw data. In these cases, the metadata were collected as completely as possible, enabling
future users to better track the data.

For each series, the time period and resolution, variables measured, location, observer, and sources of any information are
stored in our inventory. Long series are thus often fragmented in many short, sometimes overlapping sequences. The full in-
ventory is published in the supplement; an abridged version is given in Table 2. Some basic characteristics such as the length
of the records or the start year are visualised in Figs. 3 and 4.

The number of series found – over 300 – was much larger than anticipated. While many series (e.g., those of the Grisons net-
work) were rather short, we also found several long series, as well as segments that could possibly be combined into long
series (see Fig. 3). In addition to the three long (> 200 years) series that are presently available – Geneva, Basel, Great St.
Bernard Pass – many other long series could possibly be generated such as Zurich, Schaffhausen, Aarau, Bern, St. Gallen
and others. Our inventory encompasses 335 entries from 206 locations, providing an estimated 3640 station years and reach-
ing back to the early 18[th] century.

Based on this information, the data to be photographed and digitised were prioritised. Long series had a higher priority than
short series, and series from the 18[th] century had higher priority than those from the 19[th] century. The digitised data, along
with a description of the quality assurance, will be described in a subsequent paper. The focus here is on the inventoried and
photographed series, as they provide rich additional information (e.g., weather descriptions, rainfall, wind, etc.) that are not
digitised, but can be accessed online.

**4. Examples**

*The cold surge in December 1788*

In Switzerland, the winter 1788/89 was one of the coldest of the last 300 years. The winter began early, with below freezing
temperatures in the lowlands starting from 22 November. Apart from a brief warming around Christmas (a period of stormy
weather), temperatures remained low until 8 January. In the CAP7 weather type classification of Schwander et al. (2017),
which, based on station data, extends the MeteoSwiss CAP9 weather types back into the past (Weusthoff, 2011), most of the
days during this period were of type 6 ("North"), most others were type 1 ("North East, indifferent"), type 4 ("East, indiffer-
ent") or type 5 ("High pressure"). These types indicate the flow of cold, continental air towards Switzerland. Over Christ-
mas, winds turned south westerly ("West Cylconic" types 7), then back to northerly and to easterly.

As a side note, this cold winter (together with a preceding drought and the following flood-rich spring) worsened the eco-
nomic crisis (increased costs) and supply situation in France. Without overemphasising the role of weather, it arguably did
play a role in paving the ground for the French revolution in the following summer.

During this period, the Palatina network was active, providing data from two Swiss stations (Geneva and St. Gotthard Pass at
2091 m a.s.l.). Additional data were photographed and digitised from four stations: Sutz, Bern (two series for Jan. 1789),
Basel, and a second independent series from Geneva. Figure 5 shows excerpts of two data sheets. According to our invent-
ory, further data might be available for Chur, Glarus, Winterthur and Zurich, as well as from additional observers from Basel
and Geneva. However, we have not yet entirely located these data.

Raw temperature series from digitised available locations are shown in Fig. 6, together with the daily weather types. Tem-
perature series agree very well with each other over this period. All mutual correlations are between 0.89 and 0.99 except for





St. Gotthard Pass, which often lies above the inversion layer that covers the Swiss Plateau and therefore shows a clearly dif-
ferent behaviour. All lowland series show a cooling, with negative temperatures from late November onwards. During the
very cold phases, temperatures at St. Gotthard Pass were not far below those in the Swiss Plateau, which indicates stable
conditions or an inversion. Conversely, the gradient is larger during phases of cyclonic weather (e.g., type 7), i.e., the Christ-
mas warming was clearly less pronounced at St. Gotthard Pass than in the lowlands. The change from northerly to south-
westerly winds and back in late December can clearly be seen in the observations from Bern (Fig. 5), while at St. Gotthard
Pass (possibly due to channelling, which remains to be studied), winds remained north-westerly throughout the second half
of December.

This brief example demonstrates that daily weather can indeed be reconstructed from the combination of all digitised re-
cords. Adding pressure information and records from neighbouring countries could possibly allow a more detailed view at
late 18th century daily weather in Central Europe.


*The heavy precipitation event on 4 and 5 July 1817*

Precipitation measurements are sparse in the early 19th century. Although we did not systematically collect non-measured
precipitation information, the descriptions in our photographed data sheets reveal interesting insights. An example is the
flood event in July 1817, when Lake Constance reached the highest level since the beginning of measurements (see Rössler
and Brönnimann, 2018). Though melting snow contributed (at high altitudes consisting of a triple snow pack, with winter
snow from 1815/1816, which could not melt during the cold summer of 1816 but was covered by summer snow, on top of
which lay the winter 1816/1817 snow pack), it is very unlikely to have been a main cause, according to model simulations
(Rössler and Brönnimann, 2018). Rather, a heavy precipitation event in early July must have triggered the flood.

The only available precipitation series in Switzerland is from Geneva and indicates 29 mm on 5 July. This is clearly not suf-
ficient to judge whether a heavy precipitation event affected Switzerland. However, using not just the measurements but also
the weather observations and comments (which were not the focus of the project and hence not systematically collected),
more can indeed be said. Our photographed data sheets indicate rain all day long in Aarau on 4 and 5 July, and in St. Gallen
on 5 July. The monastery of Einsiedeln (see Fig. 7) noted strong precipitation already on 1 and 2 July ("pluvia, serenum"),
two thunderstorms on 4 July ("vespere tempestas, noctu repetita"), and again rain on 5 July ("pluvia, pluvia"). The observer
at Schaffhausen also noted strong thunderstorms on 4 July ("trübe, starkes Gewitter, Regen") and rain on 5 July ("fast be-
ständig starker Regen"), the notes from Marschlins concerning early July (with no clear date) indicate terrible weather with
many thunderstorms ("furchtbare Witterung mit viel Gewitter"), Vevey noted thunderstorms with hail/rain ("Gewitter mit
Hagel/Regen"), and Bern had rain on 4 and 5 July, with thunderstorms.

All sites indicate a drop in pressure and also temperature. The wind was mainly from the West or Southwest. Existing
weather type classifications indicate that 4 and 5 July were of the "Low Pressure" type (in Geneva; Auchmann et al., 2012)
or "West Cyclonic" (in CAP7; Schwander et al., 2017).

All this additional information hints at a passage of a synoptic scale system (with already antecedent rain and arguably satur-
ated soils), which was accompanied by large thunderstorm activity (incl. hail) due to embedded convection. This fits very
well with several of the weather patterns identified to lead to heavy precipitation events (Stucki et al., 2012). From this evid-
ence we can conclude that it is not unlikely that a heavy precipitation event has affected much of the Swiss pre-alpine region
and thus could have acted as a trigger for the 1817 flooding. The examples demonstrate the usefulness of the photographed
sheets; information that can then be combined with the information from historical databases such as EURO-CLIMHIST.



### 5. Conclusions

The climate of the late 18[th] and 19[th] centuries holds specific periods that are of strong interest to present-day climate science.
Studying these events requires instrumental data; however, the Swiss measurements before 1864 have never been systematically explored. The earliest Swiss measurements go back to Enlightenment scientists. The observers belonged to the liberal elite, some were members of clergy, and they were typically well embedded in a European network of scientists. Two Swiss stations were part of the Palatina network. From the mid-18[th] century onwards and throughout the first half of the 19[th] century, numerous regional meteorological networks were founded, but they all remained transitory. Only with the support from
the Swiss Federal State a national network could finally be realised (on the importance of such context-knowledge for science and applications, see Brönnimann and Wintzer, 2019).

In our study we have compiled information on more than 300 station records from Switzerland prior to 1864. The inventory is added to the global registry of the Copernicus Data Rescue Services (Brönnimann et al., 2018b), and the images are published at the open-access repository Zenodo [detailed information (URL/DOI) will follow]. Two examples illustrate the use-
fulness not only of the digitised data, but also of the images provided, for reconstructing the daily weather as far back as the 18[th] century.

Part of the data (temperature, pressure, and precipitation measurements) will be digitised and made available publicly. They can be used for various applications, including data assimilation for producing dynamical reanalyses. The example of Switzerland also suggests that the amount of European early instrumental data might have been underestimated up to now,
probably because the focus of climatology was mostly on long series. In fact, focusing on pressure measurements for the period 1815-1817, Brugnara et al. (2015) compiled and digitised more than 50 pressure series from Europe and North America and located many more. This was sufficient for a dynamical reanalysis (Brohan et al., 2016). Taken together, this suggests that at a European scale, a 250-year daily weather reconstruction should be possible. However, this requires further data rescue efforts, such as coordinated in the Atmospheric Circulation Reconstructions over the Earth (ACRE) Initiative
(Allan et al., 2011).

**Acknowledgements:** This work has been supported by Swiss National Science Foundation projects CHIMES (169676) and RE-USE (162668) and by the European Union (H2020/ERC grant number 787574 PALAEO-RA). We thank all students who helped in digitising the historical data.

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

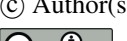



**Figures**

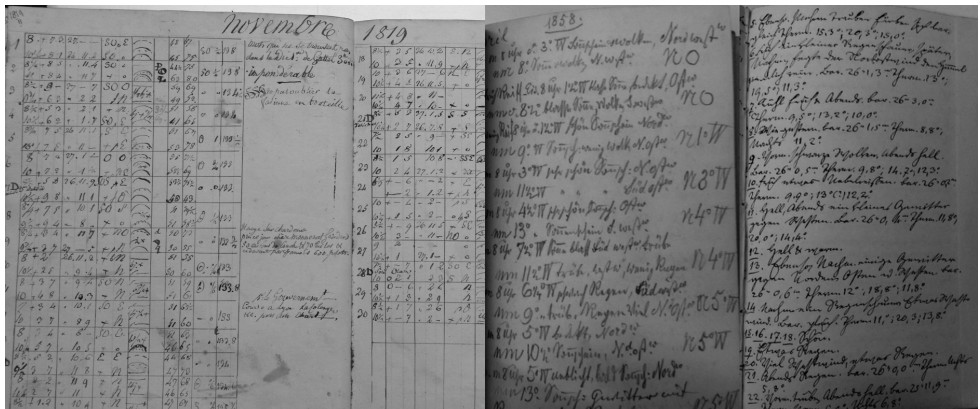

**Figure 1: Two examples of data sheets from Vevey and Schaffhausen from the Swiss Federal Archives (signatures: E3180-**
**01#2005/90#202*; E3180-01#2005/90#231*).**

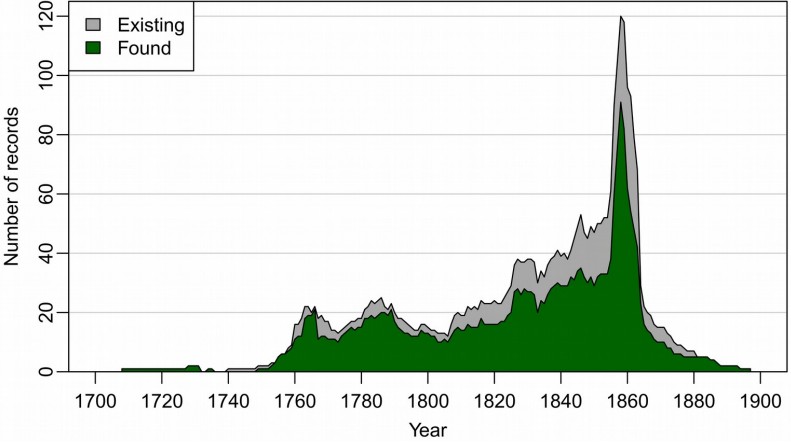

**Figure 2: Number of possibly available measurement series (grey) and number of series where subdaily data were found (green) as**
**function of the year. The peak around 1860 is due to the Brügger network in Grisons, encompassing around 90 locations. From**
**1864 onwards, only series starting prior to this date that haven't been included in the national network are depicted.**





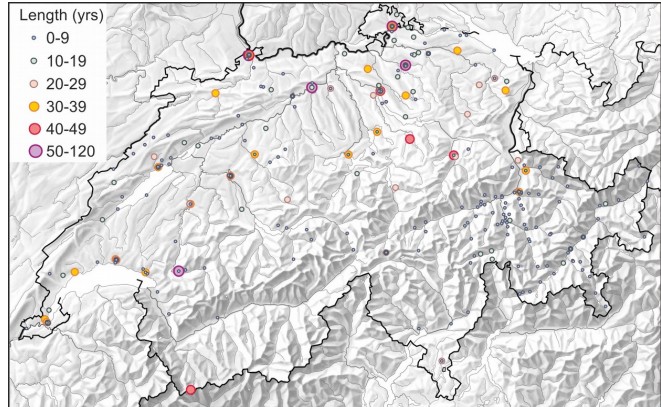

**Figure 3: Map of the available stations in Switzerland before 1864 showing the length of the series.**

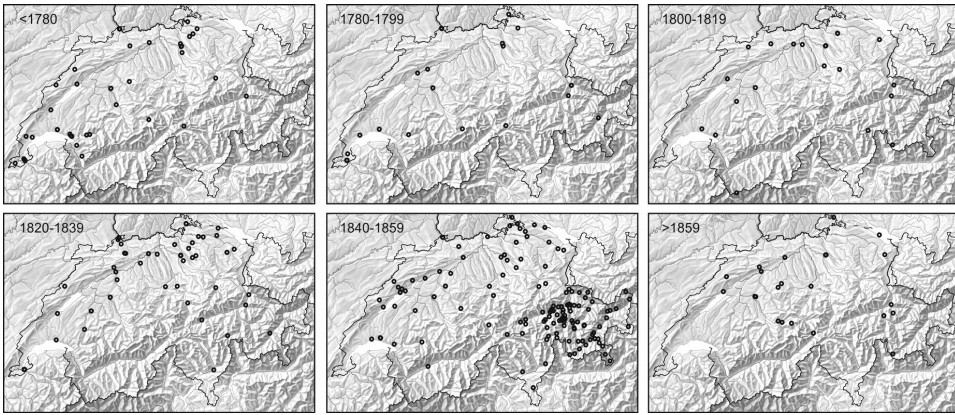

**Figure 4: Maps of the available stations as a function of the start year of the series.**



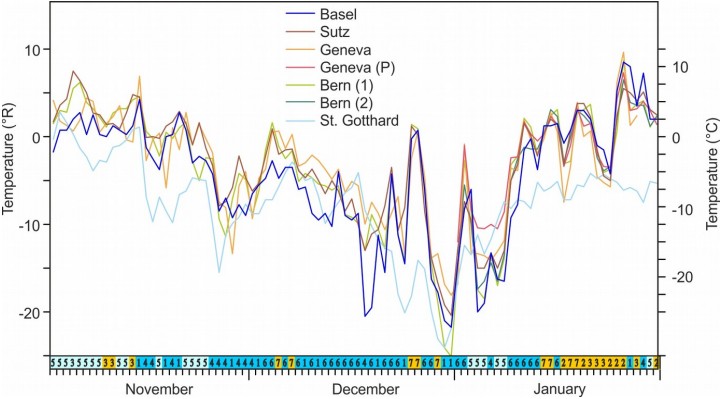

**Figure 5: Excerpts of data sheets from Bern and St. Gotthard Pass for the second half of December 1788. For St. Gotthard Pass, the wind was noted as "NW" for every single day; not shown. (Burgerbibliothek of Berne: Mss.h.h.XX.5.2; LMU Munich library:**
**0001/4 Phys. 861(1888)).**

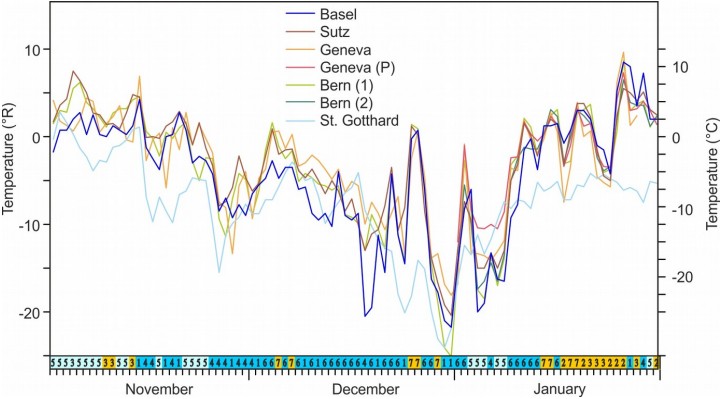

**Figure 6: Temperature at Geneva, Basel, Bern, Sutz, and St. Gotthard Pass in November to January 1788/89. If several observa-tions were made per day, we show the sunrise temperature. Note that these are the digitised raw data in degrees Reaumur. The series from Basel was shifted upward by 10° for visualisation. The bottom bar indicates the weather type according to Schwander**
**et al. (2017).**



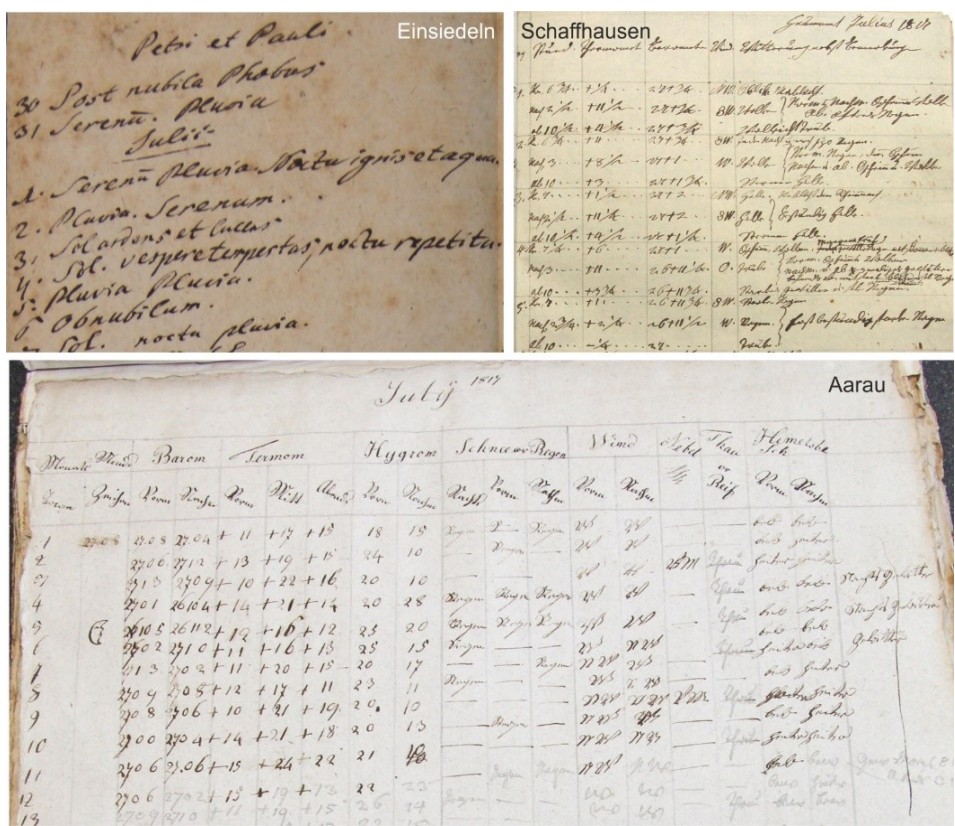

Figure 7: Excerpts of data sheets from Einsiedeln, Schaffhausen, and Aarau for early July 1817 (Klosterarchiv Einsiedeln: KAE.A.39/6; Swiss Federal Archives: E3180-01#2005/90#201*; State Archives Aargau: NL.A-0197).




**Table 1:** Meteorological networks with stations on the territory of today's Switzerland

| Duration | Organiser | Abbr. | Extension |
|---|---|---|---|
| 1760-1770 | Oekonomische Gesellschaft Bern (Bern) | ÖGB | ca. 10 stations on Bernese territory (in present-day cantons of Bern, Fribourg, Vaud) |
| 1781-1792 | Societas Meteorologica Palatina (Mannheim) | SMP | over 30 stations in European and, in a few cases, non-European regions |
| 1814-1818 | Aargauische Naturforschende Gesellschaft (Aarau) | ANG | ca. 15 corresponding stations across Europe planned, but only few observations sent in |
| 1826-1836 | Allgemeine Schweizerische Gesellschaft für die Gesammten Naturwissenschaften | SGN | 14 stations in Switzerland |
| 1855-1862 | Thurgauische Naturforschende Gesellschaft (Frauenfeld) | TNG | 5 stations in the canton of Thurgau (at the start) |
| 1856-1863 | Société des Sciences Naturelles de Neuchâtel | SSN | 6 stations in the canton of Neuchâtel (at the start) |
| 1856-1859 | Aargauische Naturforschende Gesellschaft (Aarau) | ANG | ca. 22 stations in the canton of Aargau |
| 1856-1863 | Christian Gregor Brügger (Chur) | CGB | total of ca. 91 stations in the canton of Grisons |
| 1860-1863 | Naturforschende Gesellschaft in Bern (Bern) | NGB | 8 stations in the canton of Bern |
| 1864- | Schweizerische Meteorologische Centralanstalt, Zurich (first an institution of the Schweizerische Naturforschende Gesellschaft, from 1881 onwards a Swiss federal institute) | MCH | ca. 80-100 stations in Switzerland |


**Table 2:** Summary inventory of all stations with at least 10 years of data, summarised per station (Var: p = pressure, T = temperature, w = wind, R = precipitation, day$^{-1}$ = number of measurements per day). Station names are sorted by start year of the series and indicated in the corresponding local language.


| Station | Years total | Subperiods/Observers | Comment | Var. | day$^{-1}$ |
|---|---|---|---|---|---|
| Zürich | 1708-1863 with gaps | 1708-1731 Johann Jakob Scheuchzer, 1740-1753 Jakob Gessner, 1756-1769 Johann Jakob Ott, 1759-1765 Johann Conrad Meyer, 1759-1802 Hans Caspar Hirzel, 1760-1793 Daniel von Muralt, 1781-1793 unknown, 1807-1827 Feer, 1807-1821 Hans Caspar Escher, 1812-1835 Johann Kaspar Horner, 1823-1835 H. Weiss, 1830-1832 by the Schweizerische Naturforschende Gesellschaft (F. Meyer, J.F.C. Paur, A. Nucheler), 1833-1856 J.C. Denzler, 1834-1849 Rudolf Heinrich Hofmeister, 1836-1842 by the Naturforschende Gesellschaft Zürich; (Observer: Melchior Ulrich), 1842-1857 by the Naturforschende Gesellschaft Zürich (unknown observer), 1851-1852 U. Hornig/Honig/Horner, 1859-1863 Freudweiler, 1860-1863 J. Dändliker, 1860-1866 Christian Gregor Brügger, 1861-1863 Ernst | also in nearby Wipkingen 1782-1797 by (Hans) Jakob Escher (vom Luchs) | p,T,R,w | 1-10 |
| Chur | 1749-1863 with several long gaps | 1749-1756 Johann Heinrich Lambert, 1785-1790 J.R. von Salis, 1807-1816 Johann Ulrich von Salis (Seewis), 1826-1832 Christian Tester, 1833-1840 Baroni, 1841-1843 J. Meyer, 1846-1847 unknown, 1849-1857 Leonhard Herold, 1851-1880 Major A. Buol, 1854-1856 unknown, 1856-1858 Christian Gregor | | p,T | 1-6 |



| | | | | | |
|---|---|---|---|---|---|
| | | Brügger, 1857-1857 Dr. Moosmann, 1857-1863 Wehrli, 1858-1858 unknown, 1858-1860 Hieronymus von Salis(-Soglio), 1862-1864 Eduard Killias, 1858-1859 Peter Thomas Steffani, 1857-1858 Joseph Anton Nigg | | | |
| Neuchâtel | 1753-1863 with 60 yr gap | 1753-1782 Frédéric Moula, 1844-1851 A. Bonjour, 1854-1863 Charles-Guillaume Kopp, Louis Favre | also early short series 1734-1735 by Laurent Garcin, 1861 by Adolphe Hirsch, in nearby La Tène 1859- 1861 by Fritz Borel | p,T,w | 3 |
| Lausanne | 1754-1887 with several long gaps | 1754-1790 Jean Henri Polier de Vernand, 1760-1768 Théodore Louis Traitorrens (from December 1762 Deleuze), 1763-1772 François Verdeil, 1783-1783 François Verdeil, 1808-1809 unknown, 1841-1847 Elie-François Wartmann, 1848-1872 Jules Marguet, 1855-1858 students at the Ecole Normale, 1857-1887 Marguet? | also in nearby Ouchy 1824-1828 by Henri Delessert-Will | p,T,R, w | 1-4 |
| Basel | 1755-1863 | 1755 Friedrich Zwinger, 1755-1804 Johann Jakob d'Annone, 1766-1772 Johann Heinrich Ryhiner, 1766-1795 Werner (Wernhard) de Lachenal, 1777-1785 Daniel Wolleb, 1783-1805 Abel Socin, 1784-1829 Daniel Huber, 1786 Johann Kaspar Scholer, 1825-1827 J. R. Burckhardt, 1826-1832 Johann Jakob Fürstenberger, 1826-1863 Peter Merian, 1832-1846 Andreas Schneider, 1845-1886 Gustav Adolf Huber, 1856-1874 Franz Kaufmann, | in most versions supplemented with Mulhouse and Delémont between1804 and 1826; additional measurements probably by Peter Merian in Marchmatt 1827, Arlesheim 1828, Binningen 1829-1830 | p,T,w | 1-8 |
| Cottens | 1757-1770 | Johann Ludwig Stürler | | p,T,w, R | 3 |
| Bern | 1760-1863 | 1760-1770 Franz Jakob "Monbijou" von Tavel, 1777-1789 Karl Lombach, 1779-1789 and 1797-1827 Samuel Studer, 1785-1822 Franz Rudolf von Lerber, 1803-1834 Samuel Emmanuel Fueter, 1826-1849 Friedrich Trechsel, 1837-1853 Daniel Gottlieb Benoît, 1848-1855 Johann Rudolf Wolf, 1855-1860 Johann Rudolf Koch, 1860-1863 Johann Reinhard | | p,T,w, (R) | 1-4 |
| Genève | 1760-1863 | 1760-1789 Charles Benjamin de Lubières, 1768-1800 Guillaume-Antoine Deluc, 1774-1787 Marc-Auguste Pictet, 1773-1777 Jacques-André Mallet, Abraham? Trembley, Marc-Auguste Pictet, 1778-1788 Marc-Auguste Pictet, 1787-1791 Frédéric-Guillaume Maurice, 1782-1789 Jean Senebier, 1798-1821 Marc-Auguste Pictet and Vaucher, unknown observers: 1822-1825, 1826-1835, 1836-1863 | in other versions backward extended to 1755 using Neuchâtel; also measurements Avully in 1778-1786 by Jacques-André Mallet and in Genthod in 1789-1800 by Frédéric-Guillaume Maurice | p,T,w, (R) | 1-3 |
| Orbe | 1760-1770 | Jean Bertrand and/or Benjamin Carrard | | p,T,w, R | 3 |
| Rickenbach | 1760-1777 | Friedrich David Kitt | | p,T,w | 1-3 |
| Gurzelen | 1766-1784 | Johann Jakob Sprüngli | | p,T,w | 1-3 |
| Marthalen | 1770-1781 | Johann Jakob Toggenburger | | p,T,w | 2 |
| Glarus | 1774-1818 | Johannes Marti | a short series exists for 1855-1861 by Josua Oertli | ? | |
| Waldenburg | 1776-1790 | M. (A.?) Bavier | | p,T,w | 3 |



| | | | | | |
|---|---|---|---|---|---|
| Marschlins | 1781-1863 with gaps | 1781-1785 Johann Rudolf von Salis-Marschlins, 1790-1825 Johann Rudolf von Salis-Marschlins, 1816-1816 Karl Ulysses von Salis-Marschlins, 1839-1885 Ulysses Adalbert von Salis-Marschlins | | p,T | 1-3 |
| Passo S. Gottardo | 1781-1792 | Pater Onuphrius; Pater Laurentius Mediolanensis; Jos. Belmas de Caladray | also early series 1728-1731 by Joseph da Sessa, as well as from 1844-1863 by Rigozzi | p,T,w | 3 |
| Sutz | 1785-1802 | Johann Jakob Sprüngli | | p,T,w | 2-3 |
| Rossinière | 1792-1850 | Henchoz; nephew of pasteur Henchoz from the OekGes Bern-series (from 1834 on) | also short series of Henchoz 1765-1766 | p,T,R, w | 3 |
| Schaffhausen | 1794-1863 | 1794-1845 Johann Christoph Schalch, 1833-1867 Johann Jacob Schelling (only T) | additional series from 1836-1839 Ferdinand Ludwig Peyer, 1836-1849 Johann Conrad Laffon (only T) | p,T,w | 3-4 |
| Rolle | 1798-1831 | J. F. | | p,T, R, w | 3 |
| Delémont | 1801-1832 | François-Joseph Helg | | p,T,w | 3 |
| Vevey | 1805-1859 with 14 yr gap | 1805-1840 Nicod-Delon/ Nicod de Lom, 1855-1859 D. Doret, Insp. Forestier Albert Davall | old series 1761-1766 by Perdonnet/ G. Anet | p,T,w | 2 |
| Aarau | 1807-1865 | Heinrich and Theodor Zschokke | second series 1826-1836 by Franz Xaver Bronner | p,T | 2-3 |
| Winterthur | 1808-1864 | R. Nötzli (only T) | older short series 1775-1776 by J. J. Biedermann, overlapping series from 1808-1816 (unknown), 1836-1863 (Steiner), 1843-1853 (Sulzer), 1849-1867 (Furrer) | p,T,w | 1-3 |
| St. Gallen | 1812-1863 | 1812-1832 Daniel Meyer, 1833-1853 Naturforschende Gesellschaft St. Gallen (observer unknown), 1857-1863 various | | p,T,w | 2-4 |
| Fribourg | 1816-1861 | 1816-1847 observer unknown, 1823-1847 Jean Baptiste Wière/Wiere, 1830-1859 Joseph-Victor Daguet (only T), 1856-1861 Francois (?) Moret | | p,T,R, w | 2-3 |
| Lenzburg | 1816-1845 | 1816-1818 Aug. Kl. Müller; Karl Johann Häusler, 1839-1845 Rudolf Heinrich Hofmeister | | p,T | 1-4 |
| Einsiedeln | 1817-1863 | Bernhard Foresti, Raphael Kuhn, Pius Regli | | p,T,w | 2-9 |
| Grand St-Bernard | 1817-1863 | P. Marquis, others | | p,T,w, R | 2-5 |
| Frauenfeld | 1820-1834 | Johann Conrad Freyenmuth | also later short series 1846 (unknown, only T) and 1855-1863 (Friedrich Mann) | p,T,w | |
| Herisau | 1821-1844 | 1821-1841 Johann Ludwig Merz, 1822-1845 Johann Jakob Nef | | p,T,(R) | 3 |
| Solothurn | 1823-1831 | Franz Joseph Hugi | short later series by | p,T,w | 3 |



| | | | Friedrich August Gruner 1845 and Albert Pfähler 1861-1863 | | |
|---|---|---|---|---|---|
| Weinfelden | 1824-1862 | Hafter | | p,T,w | 1-3 |
| Bever | 1826-1863 with a 5yr gap | 1826-1832 Melchior Bovelin, 1836-1841 Bovelin?, Krättli?, 1846-1863 Johann Luzius Krättli | | p,T,w | 3 |
| Luzern | 1826-1861 with 12 yr gap | 1826-1832 Josef Ineichen, 1844-1876 Franz Xaver Schwytzer | also series from 1860-1861 by Ernst Grossbach | p,T,w | 3 |
| Lindau | 1827-1841 | Unknown | | T | |
| Ellikon an der Thur | 1831-1840 | Hans Kaspar Egg | | T | 3 |
| Dielsdorf | 1834-1863 | Schoch | | p,T,w | 1-6 |
| Nufenen | 1834-1846 | Johann Friedrich Felix | | p,T,w | 3 |
| Ennenda | 1836-1859 | Jenni | | p,T | |
| Pfäffikon | 1836-1846 | Jakob Dändliker | | T | |
| Uster | 1837-1871 | J. Trümpler | | p,T | 2 |
| Altdorf | 1838-1857 | Müller | | p,T | |
| Uetliberg | 1841-1863 | Friedrich Beyel | | p,T,w | 4 |
| Fontaines | 1843-1862 | Bernard de Gélieu | | p,T,w | 3 |
| Krummenau | 1843-1863 | Johann Daniel Rothmund | | p,T | 2-3 |
| Tegerfelden | 1843-1853 | Heinrich Cornelius Sutermeister | | T,w | 3 |
| Zug | 1843-1873 | Michael Müller? | earlier series (p,T) from the 1810s and 1820s by Franz Karl Stadlin | T | 2 |
| Lugano | 1844-1863 | V. Lombardi | also short series 1856-1859 by Giovanni Cantoni | p,T,w | 3 |
| Môtiers | 1844-1856 | Barrelet | | p,T | |
| Diesse | 1845-1856 | Lamont | | p,T | |
| Stans | 1845-1863 | Bucher | | T | |
| Olten | 1846-1863 | Theodor Munzinger | also series by Hermann Frei, 1860-1863 | p,T | |
| Stäfa | 1846-1860 | Johann Jakob Dändliker | | T | |
| Stammheim | 1846-1863 | Jakob Kradolfer | | p,T | |
| Baldenstein | 1848-1863 | Thomas Conrad (von Baldenstein) | | T | 5 |
| Lohn | 1848-1863 | Alexander Beck | | T,w | 1-2 |
| Herzogen-buchsee | 1849-1863 | Adolf Albrecht Rütimeyer | | T | |
| Trogen | 1850-1862 | Tobler? | | p,T | |





| Burgdorf | 1851-1863 | Rudolf Ludwig Fankhauser | second series 1855-1859 by Friedrich August Flückiger | p,T | 3 |
|---|---|---|---|---|---|
| Diessen-hofen | 1851-1861 | Rudolf Hanhart | | p,T,w | 2-3 |
| Splügen | 1853-1862 | Florian Camastral (at the top of the pass), G. Crottogini (at the village) | | T | 3 |
| Maienfeld | 1858-1877 | Christian Enderlin | | T,w | 3 |