# Peer review of "Swiss Early Instrumental Meteorological Measurements"

_Climate of the Past, 2019_

## Short Comment (SC1) · 3 Apr 2019

submitted to *Climate of the Past, Discussions*.

The manuscript consists of three essential parts: a text, the list of references, and at last a few Figures and 2 detailed Tables. This reviewer will make some short remarks on these three parts.

The text is very interesting and explains why the findings of the (Swiss) early instrumental meteorological observations will contribute to a better understanding of decadal variability of the weather. The manuscript is in line with the evolution of the building of historical climatological data sets where the basic elements moved from average monthly climatological data in the nineties of the past century, at a later stage to daily values and to sub-daily values at present. The latter division allows a better understanding of climatic, and also of natural climate-related events, which had pronounced impacts on environment and society.

The main subject of the paper deals with the findings of an extensive archive survey of early meteorological data in Switzerland. This search resulted to the incredible estimate of 3640 station years and reaching back to the early 18th century. This reviewer having been engaged in a similar search in his home country could hardly find a fraction of that impressive result. However, it should be noted that the political and societal conditions in the 18th and early 19th centuries in his home country were entirely different which might maybe explain the lack of observational meteorological time-series. Over the last 20 years this reviewer could only add a few 19th century discoveries of meteorological data sets which remained unknown under searches carried out by scientists interested in historical climatology. Therefore, sincere congratulations to the authors of this manuscript.

The section on the history of meteorological measurements in Switzerland is very interesting and explains how Swiss scientists in the wake of the Enlightenment quantifying ideas made weather observations using the instrumental devises of their time. Furthermore, the authors describe the attempts to install a networks with the aim to publish compilations of meteorological data at different national, and transboundary, locations. The history of the '*Societas Meteorologica Palatina*', the publication of the '*Ephemerides*' comprising the meteorological information of the transcontinental network is described into some detail as 2 stations of the network were located in Switzerland.

The inventory resulted from the archive search of Swiss early meteorological measurements is well documented in the 2 tables. The text ends with 2 examples which were studied on the basis of the collected information.

Concerning the text part of the paper, this reviewer has some minor 'cons'.

It would be interesting to provide similar information on the initiating of meteorological observations by MeteoSwiss replacing the words "*and later MeteoSwiss*" (see lines 41 and 42). Of course, information on the founding of MeteoSwiss is given in lines 208 to 217.

This reviewer has the impression on the present and future availability of the digitized data sets mentioned in the manuscript. It looks like the authors remain vague and elusive on the subject.

line 53: "This made many Swiss records easier to access". How?

lines 70 and 71: "A subsequent paper will describe the digitized records"

lines 256 and 257: "The digitized data, …, will be described in a subsequent paper." Maybe this can be made more clear in the next paper.

Phenological observations has also been a source of meteorological interest. Maybe, recent research papers have probably been published on historical phenological observations in Switzerland linking them to meteorological observations and observers.

The first example dealing with the cold surge of December 1788 was not only restricted to Switzerland but encompassed a much larger part of Europe. Maybe a sentence telling this would enhance the example.

This reviewer concludes that this manuscript is excellent, its content deals with the domain of the journal '*Clim. Past Discuss.*' and therefore suggests consequently publication in the journal.

Gaston R. Demarée, Consultant, Royal Meteorological Institute of Belgium

---

## Referee Comment (RC1) · Alexander Sterin (Referee) · 5 Apr 2019

The impression of the reviewed paper is highly positive. We live and act in the era of "Big Data" (Dig - means big volume, velocity, variety - the three V's!). However the reviewed paper : Swiss Early Instrumental Meteorological Measurements - gives us an excellent example that just in the era of Big Data each part of "Small Data" (just a separate digit!!) is of high value and provides our understanding of processes in the past. The paper of Swiss colleagues from Bern demonstrates how accurate and cautious we should be with the content of old weather archives.

---

## Referee Comment (RC2) · Alba Gilabert Gallart (Referee) · 12 Apr 2019

Review to the paper: cp-2019-26

**General impression:**

Swiss Early Instrumental Meteorological Measurements. It's a very interesting paper! I really have the most positive things to say about this article. It is heading out for the science and innovative. Given these considerations and considering that the subject matter is clearly within journal scope I would recommend acceptance of this paper. I have only some small things to comment. Otherwise this is a very interesting focus.

**Quality assessment:**

Scientific significance: The paper has an impact on the field. It has a high significance in this scientific field (climatological data rescue) and is within journal scope (1).

Scientific quality: It is scientifically correct and robust. The scientific arguments and interpretation accurate and consistent with the work presented (1).

Presentation quality: The tables, images and supplementary information give a picture of the inventory made, but I really miss the link to the repository. Additionally, it would have been fine to add information on how to find the images or how the repository is structured (either in section 3 or with a new column in table 2) to more effectively and quickly link the paper and the repository (2).

**Presentation**

The writing is clear, concise and it is good English.

Abstract:

Brief and indicate the purpose of the work and what was done, what was found.

Figures:

The figures are clear to understand and make a very good summary. Only a minor comment: it would be fine if in figure 4, if a station continued operative change the colour or size of the point.

Tables: There are fine and useful tables and the captions are informative.

**Review**

Introduction:

I think the introduction is nice. The purpose is clear. Goals and lacking in science are well illustrated.

Section 2:

In an easier way it allows the reader to figure out the characteristics of the earliest measurements and most of the comments can be extrapolated to other regions. Only one question:

- Phenological data can give complementary information to past climate conditions. Did you find data (line 199)? If so, was it catalogued?

Section 3

Fine explained with figures and tables. This is very easily readable, and information presented well balanced. Comment: consider above comment about to link repository and paper. Two questions

- The authors considered ecclesiastical records and religious orders publications? It is quite frequent, for example, to find meteorological information from third countries measured by Jesuits in their Spanish libraries.
- About metadata information. You said (or I understood) that only data sheets were photographed (l 240-241) but in some cases metadata information or incidents of this type earlier publications (according my experience) were found at the beginning or end of the publication. The person in charge looked at the entire publication, to be sure that this type of information will not be lost?

 Section 4

This section adds even more value to the work done, really well developed and clear. A minor comment:

- In line 300 - 301 you said "However, using not just the measurements but also the weather observations and comments (which were not the focus of the project and hence not systematically collected),…" As you have found, many times in very ancient observations, information, especially on precipitation, appeared qualitatively but it is very important information to analyse past extreme events like your event presented or droughts… Even so, it seems that this information will not be recovered anyway.

Conclusions / future work.

The conclusion is clearly stated and provide a complete picture of the study. It is summarizing well what has been learned and why it is interesting and useful. Nice to read that the inventory was added to a global registry and part of the data will be digitised.

References

Relevant and appropriate

Alba Gilabert Gallart

PhD Centre for Climate Change (FURV)

---

## Author Comment (AC1) · 6 Jun 2019

**Reply**

The manuscript consists of three essential parts: a text, the list of references, and at last a few Figures and 2 detailed Tables. This reviewer will make some short remarks on these three parts.

The text is very interesting and explains why the findings of the (Swiss) early instrumental meteorological observations will contribute to a better understanding of decadal variability of the weather. The manuscript is in line with the evolution of the building of historical climatological data sets where the basic elements moved from average monthly climatological data in the nineties of the past century, at a later stage to daily values and to sub-daily values at present. The latter division allows a better understanding of climatic, and also of natural climate-related events, which had pronounced impacts on environment and society.

The main subject of the paper deals with the findings of an extensive archive survey of early meteorological data in Switzerland. This search resulted to the incredible estimate of 3640 station years and reaching back to the early 18 th century. This reviewer having been engaged in a similar search in his home country could hardly find a fraction of that impressive result. However, it should be noted that the political and societal conditions in the 18 th and early 19 th centuries in his home country were entirely different which might maybe explain the lack of observational meteorological time-series. Over the last 20 years this reviewer could only add a few 19 th century discoveries of meteorological data sets which remained unknown under searches carried out by scientists interested in historical climatology. Therefore, sincere congratulations to the authors of this manuscript.

The section on the history of meteorological measurements in Switzerland is very interesting and explains how Swiss scientists in the wake of the Enlightenment quantifying ideas made weather observations using the instrumental devises of their time. Furthermore, the authors describe the attempts to install a networks with the aim to publish compilations of meteorological data at different national, and transboundary, locations. The history of the 'Societas Meteorologica Palatina', the publication of the 'Ephemerides' comprising the meteorological information of the transcontinental network is described into some detail as 2 stations of the network were located in Switzerland.

The inventory resulted from the archive search of Swiss early meteorological measurements is well documented in the 2 tables. The text ends with 2 examples which were studied on the basis of the collected information.

Thank you for this very positive feedback and your suggestions to improve the manuscript.

Concerning the text part of the paper, this reviewer has some minor 'cons'.

It would be interesting to provide similar information on the initiating of meteorological observations by MeteoSwiss replacing the words "and later MeteoSwiss" (see lines 41 and 42). Of course, information on the founding of MeteoSwiss is given in lines 208 to 217.

Thank you for this suggestion. While in the introduction (lines 41 and 42), we aim at giving an overview on previous work in digitisation of instrumental series rather than describing the initiation of the national meteorological network in Switzerland, the latter is described in section two (as you mentioned). However, we will state the development of a national network in lines 41-42 more precisely in the revised manuscript. Furthermore, we will add some more information on this national network in section two (e.g. number of stations).

This reviewer has the impression on the present and future availability of the digitized data sets mentioned in the manuscript. It looks like the authors remain vague and elusive on the subject.

line 53: "This made many Swiss records easier to access". How?
lines 70 and 71: "A subsequent paper will describe the digitized records"
lines 256 and 257: "The digitized data, ..., will be described in a subsequent paper." Maybe this can be made more clear in the next paper.

Thank you for this comment. Indeed, the manuscript does not elaborate on the matter of digitisation in detail, as its focus is put more on measurement history and archive work. Following your comment however, we will give a more precise outlook on digitisation in the revised manuscript and would like to answer the individual remarks on this topic as follows:

line 53:
By transferring the archive collections from MeteoSwiss to the Swiss Federal Archives and the State Archive of the canton of Zurich, they became publicly accessible and were included in online databases of the respective archive holdings, which facilitated our archive work considerably. We will clarify this in the revised manuscript.

lines 70-71 and 256-257:
In the context of our project, more than 50 measurement series described in the inventory have been digitised (mainly pressure, temperature and precipitation) and are currently undergoing quality control and homogenisation. The digitisation concentrated on long continuous measurement series, as well as series from the 18th century. In the revised paper, we will add a sentence giving an outlook on the digitisation process and the subsequent paper.

Phenological observations has also been a source of meteorological interest. Maybe, recent research papers have probably been published on historical phenological observations in Switzerland linking them to meteorological observations and observers.

We agree that phenological data are of great value to historical climatology. In the context of our project, we encountered a considerable number of phenological records from the 18th and 19th century in Switzerland (see author's comment to RC2). To our knowledge, these records still remain to be subject to scientific investigation.

The first example dealing with the cold surge of December 1788 was not only restricted to Switzerland but encompassed a much larger part of Europe. Maybe a sentence telling this would enhance the example.

Thank you for this suggestion. In the revised manuscript, we will add a sentence on this topic to put the weather situation in Switzerland into a broader European context.

This reviewer concludes that this manuscript is excellent, its content deals with the domain of the journal 'Clim. Past Discuss.' and therefore suggests consequently publication in the journal.

---

## Author Comment (AC2) · 6 Jun 2019

**Reply to the reviewers comments**

The impression of the reviewed paper is highly positive. We live and act in the era of "Big Data" (Dig - means big volume, velocity, variety - the three V's!). However the reviewed paper : Swiss Early Instrumental Meteorological Measurements - gives us an excellent example that just in the era of Big Data each part of "Small Data" (just a separate digit!!) is of high value and provides our understanding of processes in the past. The paper of Swiss colleagues from Bern demonstrates how accurate and cautious we should be with the content of old weather archives.

We'd like to thank the reviewer for this very positive feedback and appreciate the recognition of the important role early instrumental measurements play in historical climatology.

---

## Author Comment (AC3) · 6 Jun 2019

**Reply to the reviewers comments**

**General impression:**
Swiss Early Instrumental Meteorological Measurements. It's a very interesting paper! I really have the most positive things to say about this article. It is heading out for the science and innovative. Given these considerations and considering that the subject matter is clearly within journal scope I would recommend acceptance of this paper. I have only some small things to comment. Otherwise this is a very interesting focus.

We thank the reviewer for the positive feedback and the very helpful comments and suggestions.

**Quality assessment:**
Scientific significance: The paper has an impact on the field. It has a high significance in this scientific field (climatological data rescue) and is within journal scope (1).

Scientific quality: It is scientifically correct and robust. The scientific arguments and interpretation accurate and consistent with the work presented (1).

Presentation quality: The tables, images and supplementary information give a picture of the inventory made, but I really miss the link to the repository. Additionally, it would have been fine to add information on how to find the images or how the repository is structured (either in section 3 or with a new column in table 2) to more effectively and quickly link the paper and the repository (2).

Thank you for this comment and your suggestions. The preparation of the dataset for the repository was not completed when submitting the manuscript for review, therefore no link is provided. In the revised manuscript we will add the link, as well as a few sentences on the structure of the repository in section 3. We will make sure that the dataset is well structured and the images of individual measurement series are conveniently accessible with the information given in the revised manuscript and its supplement.

**Presentation**

The writing is clear, concise and it is good English.

Abstract:
Brief and indicate the purpose of the work and what was done, what was found.

Figures:
The figures are clear to understand and make a very good summary. Only a minor comment: it would be fine if in figure 4, if a station continued operative change the colour or size of the point.

This is an excellent suggestion; we will add still operative stations to the maps in a different color in the revised manuscript.

Tables: There are fine and useful tables and the captions are informative.

**Review**

Introduction:
I think the introduction is nice. The purpose is clear. Goals and lacking in science are well illustrated.

Thanks.

Section 2:
In an easier way it allows the reader to figure out the characteristics of the earliest measurements and most of the comments can be extrapolated to other regions. Only one question:
- Phenological data can give complementary information to past climate conditions. Did you find data (line 199)? If so, was it catalogued?

Thank you for this question. Although we came across several records of phenological data from Switzerland in the context of this project, those records were not catalogued as our focus was on instrumental measurements. Nonetheless, a considerable number of phenological records from this era would exist that could provide additional information to gathered instrumental data. We will add a sentence on this topic in the revised manuscript.

Section 3
Fine explained with figures and tables. This is very easily readable, and information presented well balanced. Comment: consider above comment about to link repository and paper. Two questions
- The authors considered ecclesiastical records and religious orders publications? It is quite frequent, for example, to find meteorological information from third countries measured by Jesuits in their Spanish libraries.

Indeed, archives and publications of ecclesiastical institutions are very valuable and important sources of historical meteorological measurement series. While we found records from two such institutions (the archives of Einsiedeln Abbey and Great St Bernard Hospice), a systematic enquiry of these institutions would have exceeded available time and resources. Nevertheless, many meteorological series made in ecclesiastical institutions were found in secular publications (S. Gottardo, part of Grand St-Bernard) or have been transferred to public archives. However, some series may still be found in ecclesiastical archives. Regarding meteorological observations from third countries, only a handful was discovered in Swiss archives. Those originate mostly from travels or stays of Swiss scientists abroad.

- About metadata information. You said (or I understood) that only data sheets were photographed (l 240-241) but in some cases metadata information or incidents of this type earlier publications (according my experience) were found at the beginning or end of the publication. The person in charge looked at the entire publication, to be sure that this type of information will not be lost?

This is an important point. Such valuable metadata is crucial for e.g. homogenisation and quality control and must not be lost. In any case, we photographed the entire documents excluding only blank pages. Also metadata that could be located from other archive sources was photographed. This is not clear from the text. In the revised manuscript, we will clarify this procedure. All metadata photographed will be provided together with the corresponding measurement series in the repository.

Section 4
This section adds even more value to the work done, really well developed and clear. A minor comment:
- In line 300 - 301 you said "However, using not just the measurements but also the weather observations and comments (which were not the focus of the project and hence not systematically collected),..." As you have found, many times in very ancient observations, information, especially on precipitation, appeared qualitatively but it is very important

information to analyse past extreme events like your event presented or droughts... Even so, it seems that this information will not be recovered anyway.

Thank you for this apposite remark. Many of the measurement series found are accompanied by some sort of qualitative information on weather conditions, e.g. precipitation, snow, clouds or thunderstorms. In the inventory provided as supplement to the manuscript, we included this information in the "variables" column as wn (weather notes), not going in to detail with the specific type. However, entirely qualitative weather diaries, that are an important source of information on past weather especially before the availability of instrumental measurements, are not inventoried.

Conclusions / future work.
The conclusion is clearly stated and provide a complete picture of the study. It is summarizing well what has been learned and why it is interesting and useful. Nice to read that the inventory was added to a global registry and part of the data will be digitised.

Thank you.

References
Relevant and appropriate